# Exploring the Antibacterial Potential of Konjac Glucomannan in Periodontitis: Animal and In Vitro Studies

**DOI:** 10.3390/medicina59101778

**Published:** 2023-10-06

**Authors:** Kartika Dhipta Lestari, Edlyn Dwiputri, Geraldi Hartono Kurniawan Tan, Benso Sulijaya, Yuniarti Soeroso, Natalina Natalina, Nadhia Anindhita Harsas, Naoki Takahashi

**Affiliations:** 1Postgraduate Program in Periodontology, Department of Periodontology, University of Indonesia, Jakarta 10430, Indonesia; kartika.dhipta01@ui.ac.id (K.D.L.); edlyn.dwiputri01@ui.ac.id (E.D.); geraldi.hartono@gmail.com (G.H.K.T.); 2Department of Periodontology, Faculty of Dentistry, University of Indonesia, Jakarta 10430, Indonesia; yuniarti_22@yahoo.co.id (Y.S.); natalinahaerani@yahoo.com (N.N.); drgnadhia@gmail.com (N.A.H.); 3Division of Periodontology, Graduate School of Medical and Dental Sciences, Niigata University, Niigata 951-8514, Japan; takahashi-n@dent.niigata-u.ac.jp

**Keywords:** periodontitis, konjac, *P. gingivalis*

## Abstract

*Background and Objectives*: Periodontitis is an inflammatory disease in the supporting tissues of the teeth caused by specific microorganisms or groups of microorganisms. *P. gingivalis* bacterium is the keystone pathogen in periodontitis, so even at low concentrations, it has a considerable influence on the oral community. Antimicrobials and antiplaque agents can be used as adjunctive therapy for periodontitis treatment. Konjac glucomannan (KGM), as a natural polysaccharide, has flavonoid (3,5-diacetyltambulin) and triterpenoids (ambylon) compounds that show antibacterial activity. This research aims to analyze the antibacterial activity of KGM on animal and in vitro periodontitis models. *Materials and Methods*: The animal study divided 48 mice into four groups (control, KGM, periodontitis, KGM + periodontitis). Mice were given an intervention substance by oral gavage from day 1 to day 14, periodontitis was induced on day 7, and decapitation was performed on day 14. Samples from the right maxillary jaw of mice were used for histological preparations and morphometrics analysis. In vitro studies were carried out by adding several concentrations of KGM (25, 50, and 100 μg/mL) into a planktonic *P. gingivalis* and *P. gingivalis* biofilm. *Results*: In the animal model, KGM could prevent alveolar bone loss in the periodontitis mice model, both in histologic and morphometrics assessments. In vitro, KGM had antibacterial activity against *P. gingivalis* with better bacteriostatic (15–23%) than bactericidal (11–20%) ability, proven by its ability to inhibit *P. gingivalis* proliferation. *Conclusions*: KGM can be considered to have the potential as an antibacterial agent to prevent periodontitis. The prevention of periodontitis may improve patient well-being and human quality of life.

## 1. Introduction

Periodontitis is defined as a multifactorial inflammatory disease of the supporting tissues of the teeth that is caused by a pathogenic shift in the oral microbiome, which results in the progressive destruction of the periodontal ligament and alveolar bone with the formation of periodontal pockets, gingival recession, or both [1,2]. Periodontal disease affects about 20–50% of the population worldwide. Periodontitis results from the interaction of bacterial infection and the host response to the bacteria; the environment modifies the condition, acquired risk factors, and genetic susceptibility [3]. Multiple factors may contribute to its development. These factors include the presence of specific microorganisms. In the presence of individual susceptibility and a favorable microenvironment, these microorganisms can induce the onset of the disease [4]. Numerous works of research have demonstrated the various conditions that can influence gingival inflammation. Systemic illnesses, fluctuations in steroid hormones, dietary deficiencies, drug use, diabetes, tobacco use, and other disorders have extensive and profound impacts on the host, increasing the reaction to the buildup of bacterial plaque [5].

The most common anaerobic Gram-negative bacteria in the subgingival area are *Aggregatibacter actinomycetemcomitans* (*A. actinomycetemcomitans*), *Porphyromonas gingivalis* (*P. gingivalis*), *Prevotella intermedia* (*P. intermedia*), and *Tannerella forsythia* (*T. forsythia*). These bacteria are essential in the onset and development of periodontitis, forming periodontal pockets, the destruction of connective tissue, and alveolar bone resorption through immunopathogenic mechanisms [6].

Various studies have identified *P. gingivalis* as a keystone pathogen in periodontitis, meaning that this species [7,8,9], even in low concentrations, significantly influences the microbial community [9]. In addition to modulating the host response, the potential keystone pathogen of *P. gingivalis* can also be mediated through cell-to-cell interactions with other microbial community members [7]. *P. gingivalis* can induce oral microbial dysbiosis and increase the proliferation of various bacterial species by forming biofilms on teeth, exacerbating tooth decay [10].

In general, the host inflammatory response leads to the clearance of pathogenic bacteria. Still, in the case of periodontal disease, the host inflammatory response promotes a shift in the microenvironment that favors the growth of keystone pathogens in the tissue surrounding the periodontal pocket [8,11]. The host response to biofilm accumulation can be a double-edged sword [12]. Factors related to the host, such as genetic predisposition, immune response, and systemic health conditions, play a significant role in the occurrence and advancement of periodontitis [13]. Periodontitis has been linked to various systemic diseases, indicating a potential correlation between oral health and overall well-being. Conditions such as cardiovascular disease, diabetes, and rheumatoid arthritis have shown associations with periodontitis in research studies. Furthermore, evidence suggests that periodontal treatment, such as scaling and root planning, can benefit some diseases. For instance, it has been observed that periodontal therapy can improve glycemic control in individuals with diabetes and reduce inflammation related to cardiovascular disease [14].

The treatment of periodontitis involves good oral hygiene and professional periodontal debridement, or in some cases, requires antibiotics and periodontal surgery [15]. Plaque is the precipitating factor of periodontitis. Mechanical debridement through the adequate control of dental plaque and scaling root planning (SRP) is needed to prevent periodontitis by preventing the disruption of microbial homeostasis (dysbiosis). SRP, a more involved operation, is used on individuals who develop periodontitis. This serves as the “gold standard” of initial therapy for periodontal disease by mechanically removing plaque and calculus down to the affected teeth’s roots [16]. Antimicrobial/antibacterial agents can be used as an adjunct to mechanical debridement (scaling and root planing) by reducing oral biofilm formation without affecting the balance of the oral cavity, which is inhabited by around 1000 different bacterial species, namely 10^8^–10^9^ bacteria per mg of dental plaque [17]. Antiseptic agents, such as chlorhexidine gluconate, may give undesirable side effects, some of which are staining, dry mouth (xerostomia), altered taste sensation (hypogeusia), coated tongue, burning sensations (glossodynia), desquamation of the oral mucosa, swelling of the parotid gland and oral paraesthesia [18]. There are drawbacks associated with using antibiotics in periodontal treatment, such as the risk of antibiotic resistance, the disturbance of the natural oral microbiota, and adverse effects [19]. Thus, alternatives are needed to minimize the undesirable effects of treating and preventing oral diseases.

In recent years, konjac glucomannan (KGM) has attracted much attention due to its non-hazardous and non-toxic properties, good biocompatibility, biodegradability, and hydrophilic abilities [20]. Konjac has been widely consumed in Japan, China, and Southeast Asia as food and medicine [20,21]. Glucomannan is the main active ingredient in konjac, forming a film around food and thereby inhibiting the absorption of cholesterol and bile acids. Glucomannan also includes a protective layer on the intestinal wall and extends the stay of food in the stomach [22].

KGM is a neutral hydrophilic colloid: a natural polysaccharide derived from the rhizome of Amorphophallus konjac [23]. KGM is known to control several functions of people and human organ systems, so it is used in health food supplements, which function with anti-inflammatory, anti-cholesterol, anti-obesity, anti-diabetic, anti-cancer, antioxidant, and antibacterial properties [20,24,25,26,27,28,29]. Konjac can form an antibacterial coating on foods made from konjac flour. Flavonoid compounds (3,5-diacetyltambulin) and triterpenoids (ambylon) from konjac show antibacterial activity [22]. In the food industry, konjac is recognized as safe by the Food and Drug Administration (FDA) and can be used as a thickener, emulsifier, or food supplement. KGM-based gels are considered to have the potential for drug delivery [30,31,32]. Xiao et al. incorporate a carboxymethyl group in the KGM structure to reduce water adsorption and the molecule’s hydrophilicity. This modified structure has several promising applications for manufacturing biodegradable films, encapsulation biomaterials, and drug carriers [33].

Previous studies have proven the antibacterial effect of konjac on *E. coli*, *P. aeruginosa*, *S. aureus*, and *S. Mutans* [34,35,36]. Until now, there have been no studies that have measured the effect of KGM on periodontal tissue, including its antibacterial effect on *P. gingivalis* in the periodontitis model. Although in vitro studies can be used to study the physiological processes involved in the pathogenesis of periodontitis, the complex host responses that are fundamentally responsible for this disease cannot be reproduced in vitro.

Studies have shown that diet significantly impacts the typical oral dysbiosis associated with periodontal disease because it gives microorganisms nutritional substrates, encourages the development of a microenvironment that is favorable for the survival and multiplication of specific periodontal pathogen bacteria, and inhibits the growth of other microorganisms [37]. The main objective of this study was to assess the potential antibacterial effect of konjac glucomannan (KGM) as a preventive therapy for periodontitis using a periodontitis model. Recognizing the complex nature of this dysbiosis-based disease, this research aimed to combine in vitro trials with well-controlled animal trials. Thus, this study explored the antibacterial properties of KGM and its impact on bacterial growth in mice, aiming to gain insights into the host’s response and its influence on periodontal tissue.

## 2. Materials and Methods

This research passed an ethical review by the Research Ethics Commission of the Faculty of Medicine, University of Indonesia (KET-163/UN2.F1/ETIK/PPM.00.02/2022), Salemba Jakarta, in February 2022. The animal study used a periodontitis model of 48 male Swiss Webster strain mice aged 7–8 weeks by measuring alveolar bone destruction with histology and morphometric analysis. Mice were evenly divided into four groups, including the control, KGM, periodontitis, and KGM + periodontitis groups. This experiment was performed twice before we conducted a pilot study to ensure that KGM was safe for mice consumption and that the periodontitis model could be repeated.

In vitro research was conducted by calculating the optical density (OD) value. This study was performed at the Laboratory of Experimental Animals of the Health Research and Development Agency (Litbangkes), Salemba, Jakarta. KGM was obtained from Shimizu Chemical Corporation, Hiroshima-ken, Japan. Histological preparations were made and analyzed at the Histology Laboratory, Faculty of Medicine, University of Indonesia, Salemba, Jakarta. Laboratory tests were conducted at the Oral Biology Laboratory, Faculty of Dentistry, University of Indonesia, Salemba, Jakarta.

### 2.1. Preparation of Carboxymethylcellulose (CMC) Suspension

CMC (Cipta Kimia, Surakarta, Indonesia) with distilled water was made as a thickener for the control group of mice (placebo). To make a 0.5% CMC suspension, 500 mg of CMC powder was dissolved in 100 mL of distilled water using a magnetic stirrer until a homogeneous suspension was obtained. Stirring was carried out at 60 °C.

### 2.2. Preparation of KGM

In powder form, KGM (Shimizu Chemical Corporation, Hiroshima-ken, Japan) was mixed with distilled water at 80 mg/kg BW for the experiment [24].

### 2.3. Preparation of Experimental Animals

Forty-eight male Swiss Webster mice aged 7–8 weeks were obtained from Litbangkes (Health Research and Development Agency), Jakarta, Indonesia. Mice were evenly grouped into four groups (control, KGM, periodontitis, KGM + periodontitis) and placed in cages at room temperature with access to feed and sterile water during the study period; they were weighed every 3 days until the end of the study (day 14). On the first 7 days, a placebo (0.5% CMC suspension) (*w*/*v*) was given via oral gavage with a feeding needle daily to the control and periodontitis groups. The KGM and KGM + periodontitis mice groups were given KGM suspension (100 μL) every day from the first day, accordingly. On the seventh day, the periodontitis group underwent peritoneal anesthesia using a mixture of 10% ketamine solution and 2% xylazine with a 2:1 ratio of 1.2 mL/kg BW [38].

Periodontitis induction was carried out using a ligature wire with a diameter of 0.010 inches. The mice were fixed in a supine position, and the oral cavity of the mice was opened using a cheek retractor made using a paper clip, and a bent #25 K-file was inserted into the proximal interdental area of the first molar and second molar of the right upper jaw of the mouse to provide space for the wire. The wire was cut 1.5 mm long and bent to form the letter ‘L’ before being inserted in the proximal interdental area between the mice’s first molars and second molars of the right upper jaw. After passing through the contact point, the wire was bent to form the letter ‘U’ to increase retention (Figure 1). Mice periodontitis models were then given 200 μL of *P. gingivalis* ATCC 33,277 suspension (10^9^ CFU/mL) using an oral gavage on the day of ligation, which continued every three days until the end of the study [39,40].

All mice were then observed and included in the study according to their group. Mice that lost their ligation due to technical error were then excluded in this study.

Mice blood serum was collected by cardiac puncture after mice were euthanized. The incision was made in the abdominal area of the mouse, continued up to the chest area, and then opened until the mouse’s heart was visible. The needle was inserted into the beating heart and aspirated to obtain a blood sample. The blood was then put into a tube and allowed to stand for 15 min, and then the blood was centrifuged at 3000 rpm for 10 min at 4 °C; the supernatant was separated and stored in a freezer at −80 °C for further analysis [41].

### 2.4. Histological and Morphometric Preparations

After taking blood samples, the maxilla of the mice was taken using surgical scissors. Samples were fixed in an Eppendorf tube using a 10% neutral buffered formalin solution. The jaw samples were then decalcified for 24 h using 10% foric acid solution, and the preparations were dried with 70%, 90%, 95%, and 99% alcohol, soaked for 10 min in a xylene solution, and embedded in paraffin. Then, a 5 µm section was made using a microtome in the mesiodistal direction stained using hematoxylin-eosin (HE) before analyzing this using a microscope and ImageJ (National Institutes of Health, Bethesda, USA) software. Alveolar bone destruction on histology preparations was analyzed by drawing the Cemento Enamel Junction (CEJ) line of the first and second molar. A line is drawn parallel to the CEJ on the alveolar bone crest (ABC), then bone destruction is measured from a distance between the two lines [39]. For the morphometric preparation, the maxilla was soaked with 10% formalin solution, then the remaining soft tissue was removed; 1% methylene blue was smeared, the samples were photographed using a macro camera (Nikon, Japan), and then calculated using ImageJ software. Alveolar bone destruction was measured by calculating the area between CEJ and ABC in the first and second molar using ImageJ software. The measurement of alveolar bone destruction was conducted by two blinded clinicians unrelated to the study. The calibration between two independent assessors showed an excellent intraclass correlation coefficient (ICC = 0.9).

### 2.5. Antibacterial Activity of P. gingivalis in vitro

*P. gingivalis* ATCC 33,277 from the stock was grown on blood agar for bacterial colony selection and confirmation. After the bacteria grew, they were transferred to a Brain Heart Infusion (BHI) broth and incubated for 24 h at 37 °C in an anaerobic condition. The bacterial culture in the broth was then spread on agar with serial dilution until a concentration of 10^10^ CFU *P. gingivalis* was obtained and confirmed by the total plate count (TPC) after 24 h of incubation at 37 °C in anaerobic conditions.

For *P. gingivalis* antibacterial activity assessment, we divided *P. gingivalis* into five groups, such as the negative control, 25 μg/mL KGM, 50 μg/mL KGM, 100 μg/mL KGM, and positive control (Azithromycin 0.5 μg/mL). Bacterial activity was measured from the OD calculation and was carried out at the baseline after 12 h and 24 h using a microplate reader with a wavelength of 450 nm [39].

For *P. gingivalis* biofilm antibacterial activity assessment, *P. gingivalis* was plated and incubated for 24 h at 37 °C in an anaerobic condition into a confirmed-established biofilm; then, KGM and Azithromycin were added into the plate. Bacterial activity was measured from the OD calculation and was carried out at the baseline after 12 h and 24 h using a microplate reader (Carl Zeiss, Jena, Germany) with a wavelength of 450 nm [39].

### 2.6. Statistical Analysis

Data were analyzed using SPSS 25 (IBM, Chicago, IL, USA) software for statistical analysis, expressed as the mean with standard deviation (SD). Multivariate analysis uses a one-way analysis of variance (ANOVA) or a comparative test depending on data distribution accordingly. Significant differences in data among groups were tested statistically. The Post Hoc Test (Multiple Comparison) was used. The statistical analysis results are presented as tables or graphics using GraphPad Prism 9 (GraphPad, San Diego, CA, USA).

## 3. Results

### 3.1. Effect of KGM on Alveolar Bone Destruction (Animal Study)

Figure 2 shows the magnitude of alveolar bone destruction among the groups of mice. The perpendicular distance between the cementoenamel junction of the first and second molars and the alveolar bone crest measured bone destruction. Histological findings showed a tendency toward less alveolar bone destruction in the periodontitis group given KGM compared to the periodontitis group. The table of alveolar bone loss measurement was compiled in the Appendix A.

The results of the ANOVA test showed a significant difference among groups (*p* < 0.0001). It was found that the average alveolar bone destruction was lower in the KGM + periodontitis compared to the periodontitis group. A comparative test between the two groups revealed that there was no significant difference (*p* > 0.05) between the control and KGM group, while there was a significant difference in periodontitis and control group (*p* < 0.01), as well as in the periodontitis and KGM + periodontitis group (*p* < 0.05) (Figure 3).

Figure 4 shows the alveolar bone destruction area among groups of mice via morphometric. The alveolar bone loss is represented by the outlined area between ABC and CEJ of the first and second molar. Similar to the histology findings, a lower alveolar bone destruction pattern in the KGM + periodontitis group is also seen here.

The statistical analysis of morphometrics showed a significant difference among groups (*p* < 0.0001). Like the histology findings, the average alveolar bone destruction was lower in the KGM + periodontitis compared to the periodontitis group. A comparative test between the two groups revealed that there was no difference (*p* > 0.05) between the control and KGM groups, while a substantial difference was seen in the periodontitis and control groups (*p* < 0.01), as well as in the periodontitis and KGM + periodontitis groups (*p* < 0.05), (Figure 5).

### 3.2. Antibacterial Activity of KGM (In Vitro)

The outcomes of the comparative test analysis suggest that the OD values of the *p. gingivalis* + KGM group were considerably lower than the negative control group at 0 h (*p* = 0.01), 12 h (*p* = 0.001), and 24 h (*p* = 0.001). A comparison among the negative control group with 25 μg/mL, 50 μg/mL, and 100 μg/mL of KGM showed that *p* values were not significant (*p* > 0.05), except between 25 μg/mL and 100 μg/mL (*p* = 0.047) KGM during the 12 h incubation period, where the OD value was lower in the 25 μg/mL KGM group. A comparison of the OD values between the positive control (AZT) and KGM groups showed a *p* > 0.05 except for the 12 h incubation period between KGM 25 μg/mL and AZT (*p* = 0.047), with a lower OD value at 25 μg/mL KGM (Figure 6). In all, the bacteriostatic effect of KGM ranged from 15% to 23%.

Comparative analysis showed that there was a significant difference (*p* < 0.01) in the OD value among the *P. gingivalis* biofilm and the *P. gingivalis* biofilm + KGM group, except for an incubation period of 0 h between *P. gingivalis* and 25 μg/mL of KGM (*p* = 0.042) and 50 μg/mL of KGM (*p* = 0.225); there was no significant difference (*p* > 0.05) among the different doses of KGM (25 μg/mL, 50 μg/mL, and 100 μg/mL). Furthermore, there was no significant difference (*p* > 0.05) between the *P. gingivalis* biofilm + KGM and the *P. gingivalis* biofilm + azithromycin groups, except for the 50 μg/mL of KGM (*p* = 0.022) during the 24 h incubation period, where the average OD value of 50 μg/mL KGM was lower (Figure 7). In all, the bactericidal effect of KGM ranged from 11% to 20%. In addition, the MIC and MBC were put in the Appendix A.

## 4. Discussion

This study aimed to investigate the antibacterial effect of konjac glucomannan (KGM) in an animal model in vitro. This animal study used histological assessment to show the occurrence of alveolar bone destruction, which was significantly distinct in mice with periodontitis models. Wire placements on the first and second maxillary molars created the periodontitis model. The study by Li et al. stated that the induction of periodontitis models using ligatures wire could produce alveolar bone destruction, as can be seen through micro-CT analysis [40]. Abe and Hajishengallis reported that alveolar bone destruction in the periodontitis mice model was caused by the accumulation of bacteria, not due to mechanical trauma from the wire ligature [42]. Wire ligatures can also provide a suitable habitat for bacteria that cause periodontitis [43]. These histological studies showed that the administration of KGM did not affect alveolar bone destruction in mice. Alveolar bone destruction was reduced when mice were pretreated with KGM before periodontitis induction. Moreover, we did a preliminary study showing no statistical body weight differences in mice given KGM. This indicates that KGM is safe for consumption and may prevent alveolar bone destruction in the periodontitis mice model.

The study by Zhai et al. found that the use of KGM as a food supplement increased the immune response of Balb/C mice with tumors and reduced the suppression of immune response in tumor conditions; thus, KGM may act as a therapeutic agent with the potential to reduce tumor growth [25]. Research on the biological properties of KGM by Chen et al. showed that combining KGM with chitosan as a scaffold could provide a suitable space for the attachment of bone mesenchymal cells. In addition, adding KGM increases the biocompatibility of chitosan materials [44].

Although no studies have examined the effects of KGM on alveolar bone, KGM has been known to have an antibacterial mechanism against *Escherichia coli* and *Staphylococcus aureus*, as demonstrated in a study by Li et al. [34]. In addition, a study conducted by Chen et al. [26] using Balb/C mice concluded that KGM had a prebiotic effect on pathogenic anaerobic bacteria in the intestine and feces, namely *bifidobacteria*, *C. perfringens*, and *E. coli.* Research related to the herbal ingredient aloe vera hydrogel conducted by Putri et al. [45] stated that aloe vera extract could increase bone thickness in Wistar rats induced by lipopolysaccharide. A study by Susanto et al. [46] noted that aloe vera could reduce the number of neutrophil cells in periodontitis-model Wistar rats, which was suspected due to the fact that the activity of aloe vera’s content contains two polysaccharides: glucomannan and acemannan. Moreover, acemannan has been clinically proven to significantly improve clinical parameters in periodontitis patients, such as more significant defect depth reduction and clinical attachment levels compared to the control group [47]. It has been demonstrated that glucomannan, a polysaccharide in konjac, and acemannan have antibacterial and antiviral properties, which activate macrophages, boost immunological functions, and expedite tissue regeneration [48].

In vitro, KGM could inhibit the proliferation of *P. gingivalis* and *P. gingivalis* biofilm. In the *P. gingivalis* biofilm group, incubation was carried out for 24 h at anaerobic conditions in the medium given by *P. gingivalis* to form an established biofilm.

The study’s results on 96 well plates containing *P. gingivalis* show that KGM in all three doses could inhibit the proliferation of *P. gingivalis*. During an incubation period of 12 h, 25 μg/mL of KGM had a lower OD value compared to 25 μg/mL of KGM and Azithromycin; therefore, it can be concluded that KGM in low doses is quite effective and has potential as an antibacterial agent. Furthermore, it was found that the most significant decrease in OD values occurred within 12 h after being given KGM. This finding may provide information regarding the timing of prescribing KGM as a preventive antibacterial agent.

In the research conducted on 96 well plates with the established biofilm of *P. gingivalis,* it was observed that all three doses of KGM effectively inhibited the proliferation of bacteria at 12 and 24 h. However, there was no immediate effect observed at 0 h for 25 μg/mL and 50 μg/mL of KGM, suggesting that higher doses of KGM are required to inhibit bacterial growth in mature colonies within a short time. This is evident from the reduction in OD values for the *P. gingivalis* biofilm when compared to the 12 and 24 h incubation periods. Among the three doses of KGM, there was no difference in the OD value. KGM has the potential as an antibacterial with good ability, as indicated by the lower mean OD value compared to azithromycin at a dose of 50 μg/mL within 24 h of incubation. The results of this study suggest that the most significant decrease in OD values occurred 24 h after being given KGM; this finding can be considered when giving KGM as a curative antibacterial agent.

These findings suggest that KGM has superior antibacterial effects against planktonic *P. gingivalis* compared to *P. gingivalis* biofilms, demonstrated by a higher significance value on *P. gingivalis* than *P. gingivalis* biofilms; hence, KGM has greater potential as a preventive therapy. Previous research by Song et al. [48] found that kojic acid combined with KGM produces bacteriostatic properties and has a more significant antibacterial effect against Gram-negative bacteria than Gram-positive bacteria. Xu et al. [49] also stated that a mixture of KGM and black tea powder had bacteriostatic ability against *Staphylococcus aureus* and *Escherichia coli*. The lower OD value of the reduction in biofilms could also be due to biofilm-mediated resistance. According to Gerits et al. [50], biofilms can generate resistance associated with the impaired penetration of antimicrobials through the matrix, including the increased expression of drug-resistant genes and the decreased metabolic activity of cells residing in biofilms. Regarding this mechanism, it is speculated that KGM is a natural polymer with a hydrophilic group that makes it easy to form gels and films at a high viscosity. Though its characteristic, the strategy of hydrophilic–hydrophobic KGM may be effective against the outer membrane of *P. gingivalis*, a Gram-negative bacterium, which is hydrophobic. This possible mechanism, like fatty acid, directly affects the integrity of the bacterial membrane, resulting in an increase in membrane fluidity and permeability, thereby leading to bacterial death [39].

A study by Tang et al. in 2021 discovered that konjac glucomannan oligosaccharides effectively reduced the production of pro-inflammatory cytokines while promoting the secretion of anti-inflammatory cytokines. This study suggests that the mechanism behind the anti-inflammatory properties of konjac glucomannan oligosaccharides involves their interaction with SIGNR1. This interaction influences macrophage polarization, shifting it toward an anti-inflammatory phenotype and thus contributing to the prevention of intestinal inflammation [51]. These findings may also help KGM to reduce alveolar bone destruction in periodontitis, which is in line with its antibacterial effect [51].

The use of natural substances and medicinal herbs to prevent gingival and periodontal diseases has become increasingly popular due to their anti-inflammatory and antioxidant properties. In the present study, KGM has shown antimicrobial modality against *P. gingivalis* both in vivo and in vitro. Natural substances like KGM may be a viable alternative to contemporary pharmaceuticals as an adjuvant to scaling and root planning procedures in the clinical setting or as a host modulation therapy (HMT) agent. As proposed by Sulijaya et al., several nutrients and natural agents that have beneficial modalities may improve periodontal homeostasis [52]. Current trends have shifted into a naturally sourced agents approach with minimum side effects, namely, herbal, melatonin, and probiotics [52]. Nevertheless, the complexity of biological systems makes it challenging to control all the variables that could impact this study and interpret results confidently. In the present study, we used only one periodontal pathogen, *P. gingivalis*. Though *P. gingivalis* is well-known as a keystone pathogen in periodontal disease, it may have an antimicrobial effect toward another periodontal pathogen. The results obtained in animal and laboratory models may not necessarily be applicable to humans due to differences in physiology and metabolism. Further study focusing on the delivery system of KGM in a clinical study is warranted.

## 5. Conclusions

This research shows the antibacterial activity of KGM in an animal and in vitro study. KGM has superior bacteriostatic properties compared to bactericidal properties and has potential use as a preventive periodontal therapy. In addition to the antibacterial function, KGM has several beneficial anti-inflammatory and antioxidant functions that could possibly support the use of KGM as an HMT agent. Considering KGM is a natural polysaccharide compound and safe, the daily use or consumption of KGM may not have a bad effect. However, further research, especially with respect to the delivery system and clinical research in periodontology, is needed to gain a more comprehensive understanding of KGM’s antibacterial effects.

## Figures and Tables

**Figure 1 medicina-59-01778-f001:**
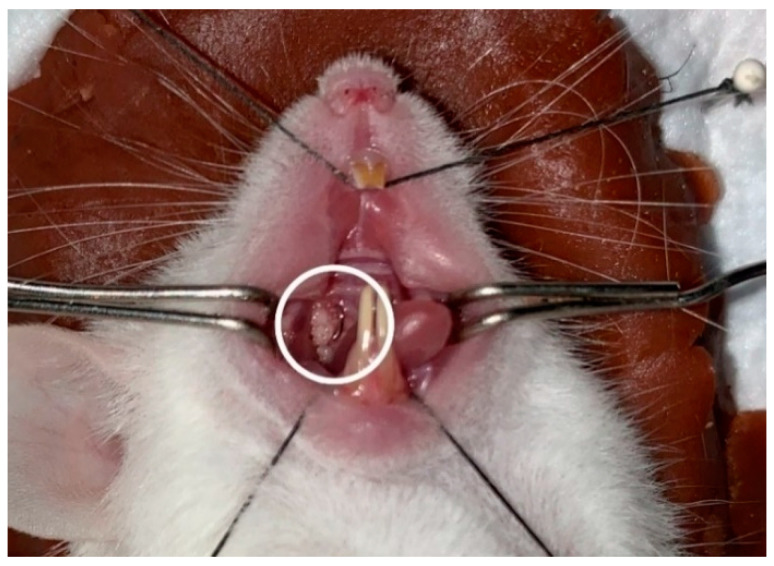
U-shaped ligature wire placement on the interproximal of first and second molar.

**Figure 2 medicina-59-01778-f002:**
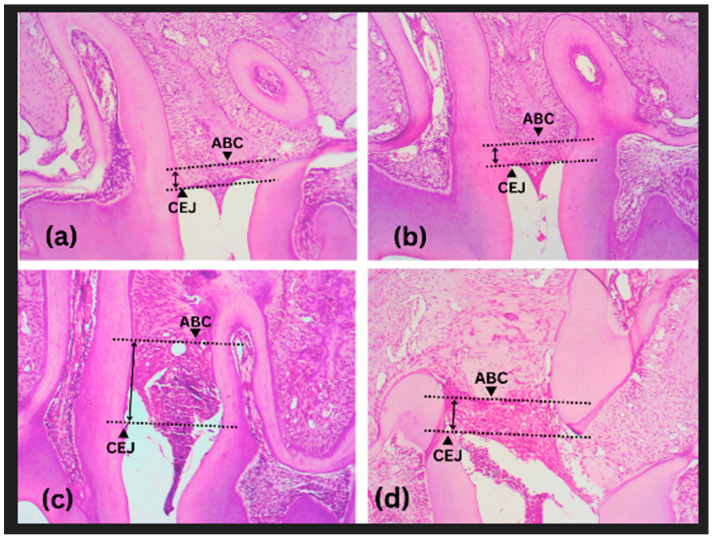
Histological findings of alveolar bone destruction (animal study). (**a**) Control, (**b**) KGM, (**c**) Periodontitis, (**d**) KGM + periodontitis; ABC: alveolar bone crest; CEJ: cementoenamel junction.

**Figure 3 medicina-59-01778-f003:**
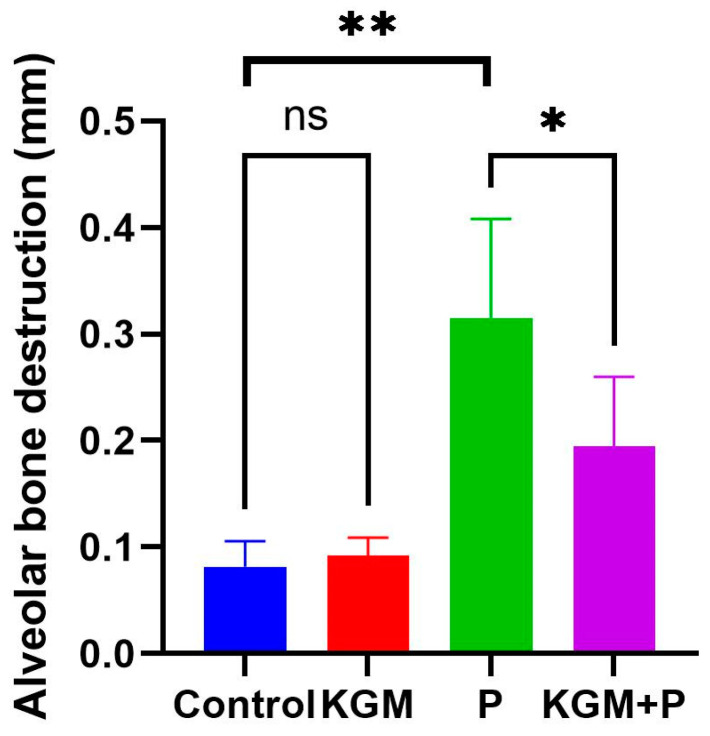
Histological evaluation of alveolar bone destruction (animal study). LSD comparative test; ** *p* < 0.05; * *p* < 0.01; ns *p* > 0.05. Control: Mice with placebo; KGM: Mice with KGM; P: Periodontitis mice model; KGM + P: Mice were treated with KGM and then periodontitis was induced.

**Figure 4 medicina-59-01778-f004:**
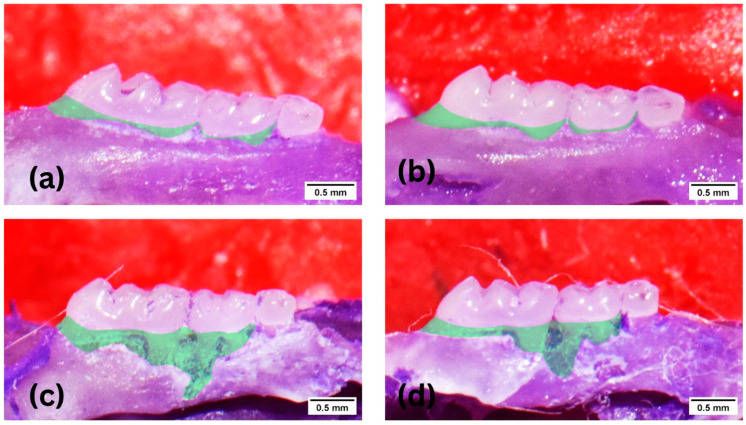
Morphometric findings of alveolar bone destruction (animal study). (**a**) Control, (**b**) KGM, (**c**) Periodontitis, (**d**) KGM + periodontitis.

**Figure 5 medicina-59-01778-f005:**
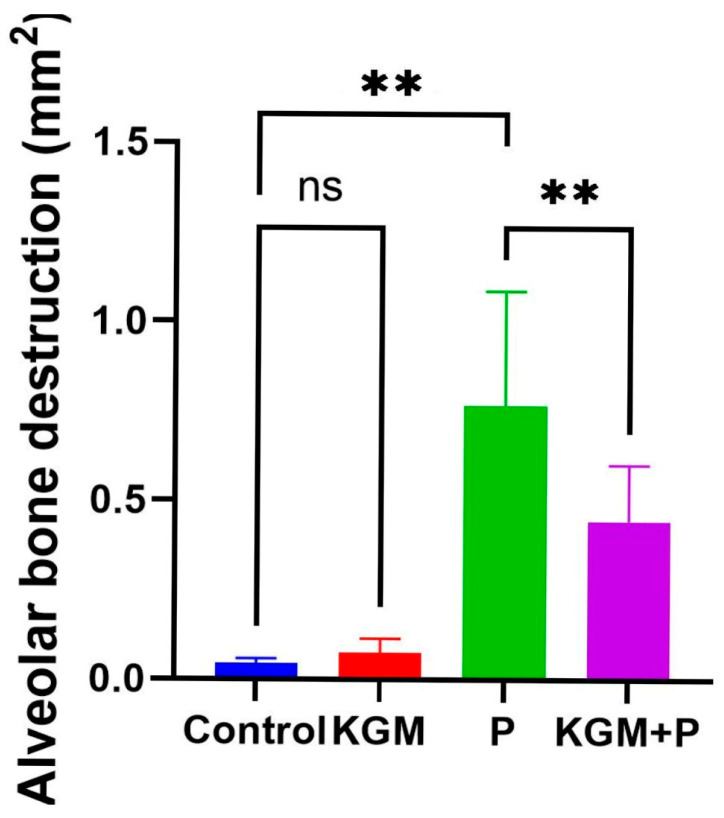
Morphometric findings of alveolar bone destruction (animal study). (**a**) Control, (**b**) KGM, (**c**) Periodontitis, (**d**) KGM + periodontitis. LSD comparative test; ** *p* < 0.05; ns *p* > 0.05. Control: Mice with placebo; KGM: Mice with KGM; P: Periodontitis mice model; KGM + P: Mice were treated with KGM and then periodontitis was induced.

**Figure 6 medicina-59-01778-f006:**
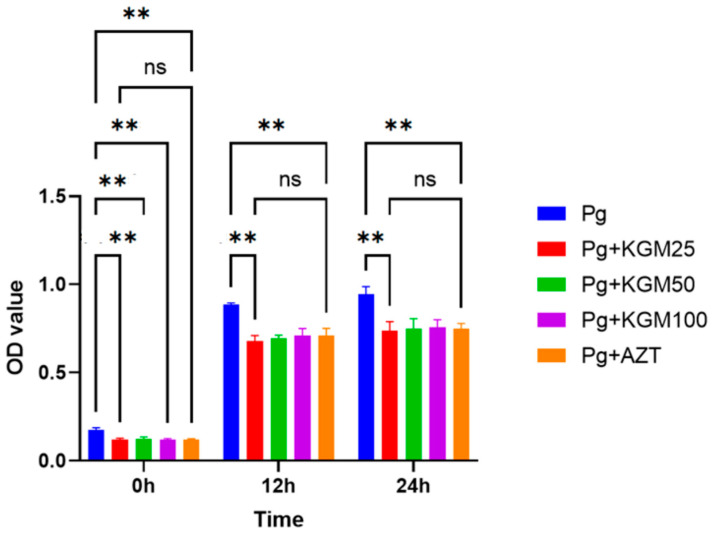
KGM antibacterial activity against *P. gingivalis* (In Vitro). Comparative test; ** *p* < 0.01; ns *p* > 0.05; Pg: *P. gingivalis;* Pg + KGM 25: *P. gingivalis* and KGM 25 μg/mL; Pg + KGM 50: *P. gingivalis* and KGM μg/mL; Pg + KGM 100 L: *P. gingivalis* and KGM100 μg/mL; Pg + AZT: *P. gingivalis* and azithromycin 0.5 μg/mL.

**Figure 7 medicina-59-01778-f007:**
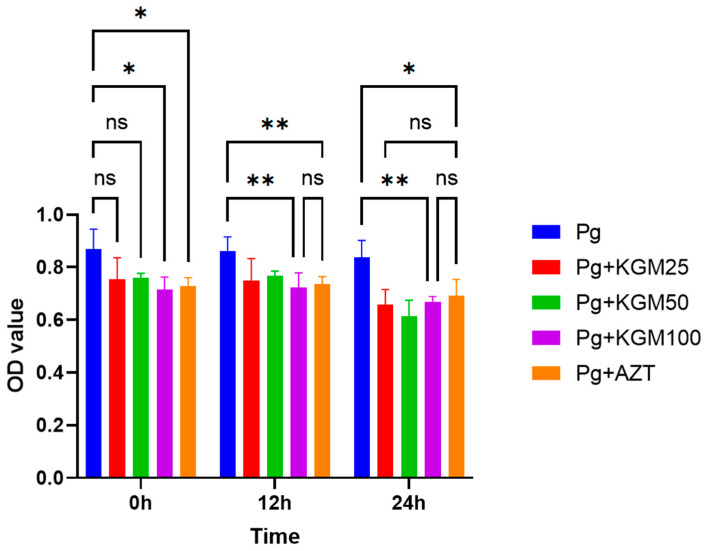
KGM antibacterial activity against *P. gingivalis* Biofilm (In Vitro). Comparative test; * *p* < 0.05; ** *p* < 0.01; ns *p* > 0.05; Pg: *P. gingivalis*; Pg + KGM 25: *P. gingivalis* and KGM 25 μg/mL; Pg + KGM 50: *P. gingivalis* and KGM μg/mL; Pg + KGM 100 L: *P. gingivalis* and KGM100 μg/mL; Pg + AZT: *P. gingivalis* and azithromycin 0.5 μg/mL.

## Data Availability

Data Availability are available upon request to the authors.

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
