# Peer review of "Exploring the Antibacterial Potential of Konjac Glucomannan in Periodontitis: Animal and In Vitro Studies"

_medicina, 2023, doi:10.3390/medicina59101778_

Round 1
Reviewer 1 Report (Previous Reviewer 2)
Dear Authors
Figure 4 was repeated.
Thank you
Author Response
We appreciate the comment given by the reviewer. We have addressed all the questions and revised it accordingly. Please find the response in the attached file.

Reviewer 2 Report (New Reviewer)
Title
The title "Antibacterial Activity of Konjac Glucomannan in Periodontitis Models: In Vivo and In Vitro Studies" is informative but somewhat lengthy and technical. It accurately describes the content of the article, but it could be more concise and engaging for a broader audience.
Title Suggestion:
"Exploring the Antibacterial Potential of Konjac Glucomannan in Periodontitis: In Vivo and In Vitro Studies"
Abstract
The abstract effectively summarizes the key points of the study, making it a useful tool for readers to quickly understand the research's scope and findings.
Comments and Suggestions to Improve the Abstract:
1. Consider adding a sentence to briefly explain why finding an effective treatment or preventive measure for periodontitis is important for public health or patient well-being.
2. The methods section of the abstract is concise but could benefit from a bit more detail regarding the specific techniques used in the in vivo and in vitro experiments. Provide a sentence or two to briefly explain the in vivo model setup (e.g., the administration method of KGM) and the in vitro experimental conditions (e.g., concentrations of KGM).
3. The results section of the abstract could be more specific about the degree of antibacterial activity observed in the in vitro studies. Include quantitative results or percentages to provide a clearer idea of KGM's effectiveness against P. gingivalis in the in vitro model.
Methods
1. Mention any specific ethical considerations or safeguards implemented during the study, if applicable.
2. It's beneficial to briefly explain why this particular mouse model was chosen and how it mimics periodontitis. Any specific references?
3. Consider briefly explaining the rationale for each group's inclusion and how they contribute to the study's objectives.
4. Add more information about the facilities or equipment used in these studies.
5. Include any specific references or protocols used for peritoneal anesthesia and periodontitis.
6. Mention the significance of measuring alveolar bone destruction and how it relates to the study's objectives.
7. Clarify why specific concentrations of KGM and Azithromycin were chosen for testing.
Results
The presentation of results is logically structured and follows the order of the methods, making it easy to correlate methods with outcomes.
Discussion
1. Discuss the potential clinical implications of the study's findings in more detail. How might KGM be incorporated into periodontal treatment strategies? Address the practicality and feasibility of using KGM in a clinical setting.
2. Elaborate on the potential mechanisms underlying KGM's antibacterial effects. For example, discuss how KGM interacts with P. gingivalis at a molecular level and how this interaction may contribute to its antibacterial properties.
3. Given that KGM has several beneficial functions, including anti-inflammatory and anti-osteoclastogenesis properties, discuss how these functions might influence periodontal health. Explore the concept of KGM as a potential host modulation therapy agent in more depth.
4. Compare KGM with existing periodontitis treatments or adjunct therapies. Highlight the potential advantages or complementary roles of KGM in treatment regimens.
Conclusion
· Conclude the discussion section by suggesting specific avenues for future research. What unanswered questions or areas of uncertainty remain? Are there potential follow-up studies that could build on the current findings? Reiterate the key takeaways from the study and emphasize the significance of the findings for the field of periodontology. Conclude with a concise summary of the main contributions of the research.
Language
The language used in this article is clear, concise, and technically sound, which greatly enhances its readability and accessibility to a wide audience.
Author Response
We appreciate the comment given by the reviewer. We have addressed all the questions and revised it accordingly. Please find the response in the attached file.

Reviewer 3 Report (New Reviewer)
Dear authors,
With interest I have read the research titled “Antibacterial Activity of Konjac Glucomannan in Periodontitis Models: Animal and In Vitro Studies”. It aimed to investigate the possible role of KGM in periodontal disease and specifically with the periodontal pathogen P. gingivalis. The research is comprehensive and provides evidence for the benefits of KGM. I would like to propose the following suggestions to be considered:
Introduction:
Line 59: “Still, in the case of periodontal disease, the host inflammatory response promotes the growth of keystone pathogens by developing inflammatory cell infiltration in the tissue surrounding the periodontal pocket” In this statement, how would the host immune response promote pathogens growth? Pathogens growth can still occur with inflammatory response through host evasion and shift in microenvironment, but these are not a real goals of local inflammatory response.
Line 76: The abbreviation of scaling and root planning (SRP) should be stated after the first mention in text.
Methods:
How were the CMC suspensions given? What was the total amount given per time?
Line 199: “The measurement of alveolar bone destruction was done by two 199 blinded clinicians unrelated to the study” What is the intraclass correlation coefficient (ICC) or agreement between the two outcome assessors?
In figure 1. Panel (a), please revise the locations of the CEJ and alveolar bone crest.
Regarding optical density, what was the Spectrophotometer device used for measuring?
What was the post-hoc test for statistically significant results of ANOVA?
Results:
Please report a table for results with alveolar bone loss values (mean +/- standard deviation), effect size and p-value for comparisons.
Discussion:
Please mention the limitations of the study such as only studying one periodontal pathogen even though oral biofilm is polymicrobial. Also, the frequency and method of administrating KMG should be investigated for optimal delivery, among others.
Conclusion:
“Anticlastogenic effect” was not assessed in the present research, please remove.
Author Response
We appreciate the comment given by the reviewer. We have addressed all the questions and revised it accordingly. Please find the response in the attached file.

Reviewer 4 Report (New Reviewer)
This is a well written manuscript.
Why (Figure 3) and (Figure 4) are erased with a line in the results section?
Please check spelling of “(Fig-ure.6)”
Author Response
We appreciate the comment given by the reviewer. We have addressed all the questions and revised it accordingly. Please find the response in the attached file.

Round 2
Reviewer 2 Report (New Reviewer)
I have reviewed the revisions made to the manuscript, and I am pleased to accept them. Thank you for your effort in addressing the feedback and making the necessary changes.
This manuscript is a resubmission of an earlier submission. The following is a list of the peer review reports and author responses from that submission.
Round 1
Reviewer 1 Report
In this paper, authors investigated the effect of konjac glucomannan (KGM) for in vivo periodontitis models and antibacterial activities of P gingivalis. The possibility of KGM to apply for the prevention of periodontitis seems novel, but the large part of methods and results in this study seems problematic. I hope my comments and suggestions help your next manuscript.
Major comments
1. L121-; about experimental animals, how many mice per each condition did you use in this study? How often did you repeat this animal experiments? Describe the detail numbers in the method section. In addition, you should explain the data usage of P. gingivalis suspension on the day of ligation and after every three days until the end of the study.
2. L165-; about antibacterial activity of P. gingivalis in vitro, at first, authors should add the culture conditions regarding atmosphare of P. gingivalis, it was the important information. Next, authors should investigate the MIC, then the results would apply for the analysis of bactericidal activity on the established biofilm. In fact, in the results of Figure 4 and 5, there were no differences at Pg+KGM50, Pg+KGM100 vs Pg+KGM25. Instead, under 25 μg/mL should be investigated. Furthermore, the methods of bactericidal activity on the established biofilm seems problematic. OD values showed just the bacterial density including both dead and alive. Additionally, usually we use the wavelength of around 600 nm to measure the bacterial density. I didn’t understand what authors want to do in this experiment. If you want to show the results of bactericidal activity using biofilm, inoculation and culture the biofilm after homogenization, live/dead analysis, or ATP measurement would be considered. Reference No.28 you cited at L177, they used the live/dead analysis. Cite the reference correctly. In addition, I recommend adding the result before 12h. Did you try to make the growth curve of P. gingivalis?
3. Figure 1; I guess this result would be the main of this study, however this histological analysis also problematic. First of all, you should add the bar in each picture to show the magnification. Next, unify the direction to make the section. Figure 1(d) was the most important results, but we could not confirm the root canals as like Figure (a) to (c). In addition, if authors want to show the alveolar bone destruction, only this histological analysis was not sufficient. Add the more graphical and quantitative analysis, like 3D CT scanning, measure the protein or mRNA expression of inflammatory cytokines in the periodontal tissue. See reference No. 27 you cited.
4. L285-298, Figure 3; this result should be related with the results of Figure 1. Why authors used this assay, you understand 14 days was short to measure P. gingivalis IgG? I recommended omitting this results, instead of the addition of evaluation the alveolar bone destruction.
Minor comments
1. Bacterial name should be italic in whole text.
2. L39; Actinobacillus actinomycetemcomitans was changed the genus, “Aggregatibacter actinomycetemcomitans.
3. Keep the rule of notation of bacteria, use the half-width space between genus and species name.
4. L42; The spelling of Tannerella forsythensis inaccurate. Correct Tannerella forsythia.
5. L120; Describe the kind of water.
6. English proof reading is required.
English proof reading is required. Check the typo including the usage of space.
Author Response
We thank the reviewer for giving us positive suggestions and corrections to improve this manuscript. We have revised the manuscript according to the reviewer's comment. The response file is attached. We hope this revised manuscript suits for publication in Medicina.
Regards,
Benso

Reviewer 2 Report
The manuscript titled "Antibacterial Activity of Konjac Glucomannan in Periodontitis Models: In vivo and In vitro Studies" concerns the efficacy of Konjac Glucomannan in periodontitis models. The subject is interesting given alternative treatment strategy to control periodontitis. However, there are several comments and suggestions which you may find below:
1. Title: "in vivo" may be misleading and replaced by "animal"
2. Abstract: Page 1 Line 4: the word bacteria is plural
3. Once abbreviated in the abstract, please use the abbreviated form (KGM) within the abstract. And this also apply throughout the main text. Please check carefully
4. P1 L16, Antimicrobials and antiplaque agents .......
5. There are numerous grammar errors, for example:
P1 L20-21
P3 L101-103
P9 L287-289
P9 L308-309
P10 L333-337
6. There are some sentences which are needed to be rephrased, for example:
P3 L106
P4 L173-175
P6 L216-219
P8 L245-248
P9 L287-289
P9 L299-302
P10 L334-337
7. P1 L23, ...mice were used for.....
8. P1 L25, not understood "..into P gingivalis and P. gingivalis biofilm"
9. P2 L55-56, replace "professional dental cleaning" with professional periodontal debridement"
10. P3 L112, Universitas ???
11. P3 L124, replace "Mice" with "They"
12. P 4 L162, ...kit were left 30 ......
13. P4 L175-177, Frthermore, OD calculations were carried out at baseline ......
14. P4 L179, data = plural
15. P4 L180, ...expressed as mean....
16. Please state the calculation for the power of the study
17. P4 L190, ...a tendency towards...
18. When more than 2 groups are compared, please use "among" instead of "between"
19. Please check spaces between the words throughout the text
20. Provide another picture for Figure 1d
21. Use “<” instead of “≤”
22. P6 L205, delete "Kruskal-Wallis comparative test was performed"
23. P6 L218, what do you mean with "P. gingivalis group" ? Negative control group maybe??
24. P8 L253-4, P9 L 276-277, cite references
25. P10 L310-315, this sentence is far too long, write in two sentences
26. P10 L 312, give effect ???
It is adviced to seek help from a native speaker.
Author Response
We thank the reviewer 2 for the valuable insight and comment to our manuscript. We have revised the manuscript as suggested. We hope now it suits the publication.
Regards,
Ben

Reviewer 3 Report
In the manuscript entitled “Antibacterial Activity of Konjac Glucomannan in Periodontitis Models: In Vivo and In Vitro Studies” the authors aimed to analyze the antibacterial activity of konjac glucomannan on in vivo and in vitro periodontitis models. The study is well conducted and presented. Here are some suggestions to improve the quality of the manuscript:
ABSTRACT
_ In the abstract, ample space is devoted to the background of the topic, but little to the description of the results and conclusions of the study.
INTRODUCTION
_ The authors correctly explained the bacterial etiology of periodontitis, hinting at the concept of dysbiosis, which should, however, be better explained.
_ The host-related factors in the manifestation of periodontitis should also be mentioned.
_ Mention how periodontitis has been associated with various systemic diseases (with respective literature citations) and how periodontal treatment may have positive effects on some of these diseases.
_ I suggest listing the undesirable effects of common antiseptics used for oral bacterial load control to better clarify the reasons for this study on konjac glucomannan (KGM) properties.
MATERIALS AND METHODS
_ If possible, it would be nice to include a clinical image of the U-shaped ligature wires in the mice’s mouths in the manuscript.
DISCUSSION
_ Start this paragraph by mentioning the aim of the study.
Author Response
We thank the reviewer 3 for the valuable insight and comment. We have revised the manuscript accordingly. We hope now this manuscript suits the publication.
Regards,
Benso

Reviewer 4 Report
In the manuscript titled "Antibacterial Activity of Konjac Glucomannan in Periodontitis Models: In Vivo and In Vitro Studies," the authors investigate the antimicrobial activity of Konjac Glucomannan in vivo and in vitro, concluding that KGM can be considered to prevent periodontitis.
Nowadays, prebiotics has become increasingly popular in the prevention and treatment of various diseases, so I think the following study is very promising and interesting. However, I think the work needs some medications in order to improve its quality.
1. Introduction
Lines 32-35
I believe that it is reductive considering current knowledge to define that periodontal disease is induced exclusively by specific microorganisms. Please better define that it is a disease with multifactorial etiology in the origin of which multiple factors contribute, including the presence of specific microorganisms that in the presence of individual susceptibility and a suitable microenvironment can induce disease.
Lines 55-56
I believe it is appropriate to better define "professional dental cleaning." Emphasize how the first-line treatment of periodontal disease consists in planning and root planning, the main purpose of which is to mechanically remove bacterial biofilm.
Lines 59-64
Specify that these are adjuvants to nonsurgical periodontal therapy (scaling and root planning). Indicate why antibiotics are poorly indicated for this purpose.
Lines 91-97
More coincidentally define the objective of the study.
Specify why the use of pre and probiotics in dysbiotic-based diseases is important.
See this manuscript “Santonocito S, Giudice A, Polizzi A, Troiano G, Merlo EM, Sclafani R, Grosso G, Isola G. A Cross-Talk between Diet and the Oral Microbiome: Balance of Nutrition on Inflammation and Immune System's Response during Periodontitis. Nutrients. 2022 Jun 11;14(12):2426. doi: 10.3390/nu14122426. PMID: 35745156; PMCID: PMC9227938”.
2. Discussion
Lines 266-277
Are there any in vivo studies on human specimens evaluating the activity of glucomannan and acemannan in the periodontitis and bone rigeneration?
See this article “Ipshita S, Kurian IG, Dileep P, Kumar S, Singh P, Pradeep AR. Agenti modulanti dell'ospite locale dell'alendronato e del gel di aloe vera in pazienti con parodontite cronica con difetti di forcazione di classe II: uno studio clinico randomizzato e controllato. J Investig Clin Dent. 2018 August; 9(3): e12334. DOI: 10.1111/jicd.12334.”.
It was observed that Konjac glucomannan can prevent intestinal inflammation through SIGNR1-mediated regulation of activated macrophages, having a regulatory action in host immunity and intestinal homeostasis. Do you think the protective action observed in the study toward periodontal disease can also be attributed to this action, considering the role of macrophages in the pathogenesis of periodontal disease? There is much focus on the activity that Konjac glucomannan has on microorganisms implicated in the development of periodontal disease, less so on its potential in modulating the host immune system.
3. Conclusion
reformulate the conclusions as extremely reductive and incomplete.
ì
Make slight form corrections
Author Response
We thank the reviewer 4 for the valuable insight and suggestion. We have revised the manuscript accordingly. We hope now it suits the publication.
Regards,
Benso

Round 2
Reviewer 1 Report
I’m disappointed with the authors revision. Authors should be more polite against your research. Authors could not enough revise the manuscript according to the comments and suggestions from reviewers. In addition, references’ citation messed up. It’s kind of violence against the science. Did all authors check the manuscript before the submission? I recommend starting over this study.
1. Regarding “antibacterial activity of P. gingivalis in vitro”, the explanation of methods was so poor. Authors must rewrite this section. I pointed out at the first round, I could not understand what authors want to investigate in these experiments.
1) Widyarman AS et al. (authors showed in author’s responses) used the typical biofilm assay with crystal violet and safranin staining to count “the biofilm mass”. Did authors use the same assay? Why didn’t you cite this reference? Even if you used the same biofilm assay, you could not measure “the bactericidal activity”. If you used the WST method to measure the biofilm, you could count the bactericidal activity with 450nm wavelengths. Anyway, it is unclear how did you make and count the biofilm.
2) About the reference Zeng Y et al (authors showed in author’s responses), the concentration of 50 to 200 μg/mL konjac oligosaccharide were adjusted for cytokine assay, not bacteria. Authors should not use this reference data for your examination. You must retry the experiments from MIC and MBC assay to set the concentration for analyzing the antibacterial activity of P. gingivalis.
3) Even If you could not do the live/dead or ATP assay, you could the general microbial procedure, “inoculation and culture the biofilm after the homogenization” I proposed at the first round. You can see the living colonies in biofilm and show the bactericidal activities.
2. Regarding revised Figure 1, each magnification from (a) to (d) (especially (c) and (d)) was obviously different. However, authors add the same length bar for each picture. I think it was incorrect data.
3. Regarding Figure 3, bars were also required. It was difficult to distinguish difference between (c) and (d) because of the intensity of the staining. Especially Figure3(c) was difficult to see the CEJ and ABC. I recommended using another picture selection.
4. I already pointed out at the first round, add the repeat number of animal experiments.
5. L206; P. gingivalis ATCC 33277T, not ATCC 33277 P. gingivalis.
I think English problem was almost corrected in the revised manuscript.
Author Response
We thank the reviewer 1 for the valuable suggestion to our manuscript. We kindly ask more time to run the MIC and MBC. Meanwhile, we submit the revised version of our manuscript. We have revised some typos, corrected figures, and double-checked the references as suggested.
Regards,

Reviewer 2 Report
The revised version of the manuscript was improved. However some minor corrections are still needed, as follows:
1. Spelling check is highly advised
2. L133-135, grammar error
3. L203, measurement
4. L204, clinicians
5. L216, OD has already been abbreviated
6. L259-261, the sentence not understood, lower than what?
7. L322, impotive ???
8. L377 therapy
Please check spelling
Author Response
We humbly thank the reviewer 2 for giving us valuable corrections. We have revised the manuscript accordingly. We hope now it suits for publication.
Regards,

Round 3
Reviewer 1 Report
See attached. My comments shows in red characters. Anyway authors must not submit the incomplete manuscript.

After rewrite the manuscript, I think it is required the English proofreading again.